# Modified mRNA as a Treatment for Myocardial Infarction

**DOI:** 10.3390/ijms24054737

**Published:** 2023-03-01

**Authors:** Yu Wang, Meiping Wu, Haidong Guo

**Affiliations:** 1Academy of Integrative Medicine, Shanghai University of Traditional Chinese Medicine, Shanghai 201203, China; 2Science and Technology Department, Shanghai University of Traditional Chinese Medicine, Shanghai 201203, China; 3Department of Anatomy, School of Basic Medicine, Shanghai University of Traditional Chinese Medicine, Shanghai 201203, China

**Keywords:** myocardial infarction, modified mRNA, myocardial regeneration, gene therapy, paracrine effect

## Abstract

Myocardial infarction (MI) is a severe disease with high mortality worldwide. However, regenerative approaches remain limited and with poor efficacy. The major difficulty during MI is the substantial loss of cardiomyocytes (CMs) with limited capacity to regenerate. As a result, for decades, researchers have been engaged in developing useful therapies for myocardial regeneration. Gene therapy is an emerging approach for promoting myocardial regeneration. Modified mRNA (modRNA) is a highly potential delivery vector for gene transfer with its properties of efficiency, non-immunogenicity, transiency, and relative safety. Here, we discuss the optimization of modRNA-based therapy, including gene modification and delivery vectors of modRNA. Moreover, the effective of modRNA in animal MI treatment is also discussed. We conclude that modRNA-based therapy with appropriate therapeutical genes can potentially treat MI by directly promoting proliferation and differentiation, inhibiting apoptosis of CMs, as well as enhancing paracrine effects in terms of promoting angiogenesis and inhibiting fibrosis in heart milieu. Finally, we summarize the current challenges of modRNA-based cardiac treatment and look forward to the future direction of such treatment for MI. Further advanced clinical trials incorporating more MI patients should be conducted in order for modRNA therapy to become practical and feasible in real-world treatment.

## 1. Introduction

According to global death statistics, cardiovascular diseases (CVDs) are the first cause of mortality [1], therein, myocardial infarction (MI) accounts for 46% of deaths in CVDs [2]. MI induces multiple complications from myocardial necrosis and fibrosis of the heart milieu to whole heart damage with limited ability to regenerate [3]. Difficulties in MI treatment still exist, despite updated medical methods that have been developed over the decades.

With the development of gene editing technology, significant progress has been made in clinical translations and applications of gene therapies, including mRNA-based therapy, DNA-based therapy, and recombinant proteins [4]. In terms of the aims of MI treatment, mRNA should be a better substitute for DNA or recombinant proteins, due to its transient expression for transcription to DNA and encoding proteins [4]. However, mRNA-based therapy still has limitations that should be further addressed. Firstly, mRNA is quickly degraded by ribonucleases (RNase) because of host defense activities [5]. Therefore, exogenous mRNA is extremely unstable when transferred to a specific organ, and there is insufficient protein translation of exogenous mRNA [6]. Secondly, exogenous mRNA has high immunogenicity, which induces a potent immune response via the activation of Toll-like receptors (i.e., TLRs, TLR7, and TLR8) [7,8,9]. Thirdly, uridine in mRNA renders translation difficult for the involvement of RNA-dependent protein kinase (PKR) with the ribosome inhibiting mRNA [10,11]. The low efficiency of translatability, instability, and high immunogenicity of exogenous mRNA are crucial limitations to be optimized for effective therapeutic application.

Due to these potential limitations, there has been significant interest in how to successfully transduce exogenous mRNA into cardiac cells. The aim of modRNA-based gene therapy is to achieve powerful protein translation with low immunogenicity, stability, as well as a low risk of insertional mutagenesis of exogenous mRNA [12]. In this review, we discuss the optimal conditions for modRNA-based therapy by using gene modification methods and selecting suitable delivery materials. Moreover, the effects and mechanisms of modRNA in MI treatment are discussed. Finally, we discuss the current challenges of modRNA-based cardiac treatment and look forward to the future direction of such treatment for MI.

## 2. Structural Basis of modRNA

Production of mRNA in eukaryotic cells involves several processes that include adding 5′ cap, splicing to delete non-coding introns, and forming the 3′ terminus [13]. The 5′ cap and untranslated regions (UTR, 5′ UTR, and 3′ UTR) have the ability to increase transcript stability and initiate translation in whole processes [14]; 5’ UTR seems to be the crucial driver of protein expression, in which distinct 5′ UTR characteristics have different effects on the mRNA translation [15,16]. For example, one study built a library of 300,000 randomized 5′ UTRs and created a potential model regarding 5′ UTR sequences and translation efficiency which allowed the design of 5′ UTRs with enhanced protein production [17]. The 5′ UTR modRNA (adding 5′ UTR with GATA2) has been shown to promote differentiation of the pluripotent stem cells (PSCs) into endothelial cells (ECs) [18]. In addition to designing 5′ UTRs, the efficiency of translation can be increased by adding 3′ UTR twice in tandem [19]. The last step of mRNA transcription is the insertion of a poly-A tail, which is located at the 3′ terminus; the long poly-A sequences facilitate nuclear transport, commence translation, and enhance mRNA stability [20].

Internal modifications on mRNA are also crucial in translational efficiency. The modified nucleosides change the secondary structure of RNA by altering hydrogen bonding patterns, influencing base stacking potential, or favoring a specific nucleotide conformation [21]. By editing nucleotides of mRNA, including insertion of methyl or hydroxylate groups and replacement of uridine with pseudouridine (ψ), the TLR signaling pathway is inactive, and thus inhibits immunogenicity [22]. Furthermore, the translation capacity of mRNA can be significantly enhanced through the replacement of uridine with ψ by inhibiting the PKR pathway [23] and RNase activity [24]. Nevertheless, mRNA modification seems to be a double-edged sword, with a context-dependent translating ability in different cells and different coding sequences [25]. To solve these limitations, several studies have been undertaken to optimize the technology for increasing the translational capacity of modRNA.

Modified mRNA is a nucleotide-modified vector, which instantly translates sufficient proteins with gradual degradation (5–7 days in vitro, 10 days in vivo) and low immunogenicity [26]. Commonly, chemically modified nucleotides, such as 5-methylcytidine (5meC), N^6^-methyladenosine (m^6^A), and N^1^-methylpseudouridine (m^1^ψ) have effective functions in mRNA translation (Figure 1) [27]. Studies have validated that both the types of fluorescent vectors [28] and the pH values [29] affect the translated capability of modRNA. Accordingly, enhanced green fluorescent protein (eGFP) mRNAs with m^1^ψ have been shown to have the most potent expression of encoded protein as compared with 5moU, ψ, and encoding firefly luciferase (FLuc) modifications [28]. This could be because of a reduction in immune response with antigen-specific cell-induced m^1^ψ modRNA [30], increased ribosome loading [31], augmented microRNA, and protein sensitivity [32]. However, eGFP coupled with 5moU has been shown to be less sensitive to RNase than other modRNA [28]. David et al. compared eGFP expression between nucleotide-modified RNA and unmodified RNA, and the results showed that mean protein expression increased 1.5-fold for m^1^Ψ modRNA in HeLa cells [33]. Interestingly, the temperature at neither 30 °C nor 37 °C affected the translated results, which means the temperature was independent of the translation [28]. Moreover, 5meC-modified mRNA had optimized protein expression at pH 5, and ψ did best at pH 7 [29].

Codon modification also influences the translation efficiency of mRNA. The “GC3” codon combined with m^1^ψ has produced an mRNA more than 1000-fold than other mRNA variants and outperformed all other unmodified mRNA [34]. Moreover, the optimal dosage of N1-methylpseudouridine-5′-triphosphate nucleotide (1-mΨU) modRNA in MI treatment was 0.013 μg modRNA/mm^2^ with RNAiMAX of cardiomyocytes (CMs) in vitro and 1.6 μg/μL with sucrose-citrate buffer in vivo [35,36]. In addition, Andries et al. coupled m^1^ψ with 5meC and found that the combination of both increased translation efficiency and reduced immunogenicity, which was superior to modifying alone [37]. Post-translational modifications have been shown to affect protein expression in various ways [38]. The glycosylation of human follistatin-like 1 (hFSTL1) significantly influenced cardiac function and transported the mutation of a single site (N180Q) hFSTL1 modRNA into the myocardium, increasing the proliferation of CMs and myocardial regeneration in MI mice [39]. Moreover, Yiangou et al. encoded Ca^2+^ indicators (GEVIs and GECIs) in modRNA and delivered them into human-induced pluripotent stem cell-derived cardiomyocytes (hiPSC-CMs) for long-term observation (lasting for 7 days), and demonstrated the value of modRNA as a gene delivery method [40].

In conclusion, the translation efficiency of modRNA can be further elevated by editing nucleotides, combining the actions of modified nucleosides, optimizing the 5′ UTR and 3′ UTR vector sequences, an optimal pH value, as well as optimal dosages in vitro and in vivo.

## 3. Delivery Vectors and Methods for modRNA in MI

### 3.1. The Viral Delivery Vectors

Traditional virus vectors, including adenovirus, adenovirus-associated virus (AAV), and lentivirus have limited insert capacity for mRNA transduction [41], whereas non-viral vectors such as naked DNA plasmid and modified mRNA (modRNA) do not have size limitations, and can support therapeutical genes regardless of size. Moreover, the modRNA method has an overwhelming characteristic that most viral vectors lack, i.e., it is not affected by the conditions of the nuclear membrane, thus, it can transfect either dividing or non-dividing cells. Furthermore, MI has its own unique temporal expression window within 3 d of the inflammation phase and significant changes can even happen in cardiac cells as early as 24 h; therefore, the modRNA is an ideal vector for optimal delivery with a specific time window for protein expression. Virus vectors, especially AAV, can prolong gene expression for up to 4 weeks, and even 11 months in MI [42]. This uncontrolled and long-term gene delivery method can miss the optimal time of treatment, and can also induce some unnecessary risks. Accordingly, the specific pulse-like, transient gene expression of modRNA is highly favorable for protecting the heart from further damage [43]. In addition to the risk from constant gene expression, adopting viruses for gene therapy poses a series of safety concerns. Lentivirus vectors are rarely used in MI with their high immunogenicity, as well as their random insertion into a host with a preference of targeting coding regions can cause risks of insertional mutagenesis and tumorigenesis [44]. Although AAV vectors without immunogenicity are highly preferred in MI treatment, a critical limitation in AAV therapy is the translation efficiency, because from 30 to 50% of the protein is neutralized by the presence of pre-existing anti-AAV antibodies [45].

### 3.2. The Non-Viral Vectors

By contrast, non-viral vectors are favorable to use as delivery vectors. Achieving the ideal biodistribution of the target gene in the myocardium is still a challenge in MI treatment. There are multiple vectors for delivering modRNA both in vivo and in vitro. Gene knockout induced via RNA interference (RNAi), a biological activity regulated through double-stranded RNA and referred to as small non-coding RNAs (20–30 nucleotides), is widely applied in MI therapy. By encapsulating modRNA with RNAiMAX (a new transfection reagent) based on the cationic lipid formulation that possesses polar heads and non-polar tails, transfection efficiency has reached the maximum in vitro [46], whereas it was not an ideal choice to inject into the myocardium [35]. The administered RNAiMAX faces several obstacles in accomplishing its targets, i.e., it needs to pass through cellular membranes as well as evade enzyme processes and immune-induced degradation.

To overcome these problems and to enhance the stability of mRNA in vivo, the application of nanoparticles is recognized as the best biomaterial to encapsulate modRNAs. Based on the biomaterial used, nanoparticles (10–1000 nm) are subgrouped into inorganic, organic (e.g., micelles, liposome, protein-based carriers, polymers, and cyclodextrins), viral, and mixed nanoparticles [47]. Delivering nanoparticles needs to avoid agglomeration and to retain them in colloidal suspension. These functions require assistance from several materials, including polyethylene glycol, dextran, chitosan, pullulan, and sodium oleate [48]. In acute MI, M3-FLuc modRNA delivered via intramyocardial injection increased protein expression in primary CMs and lasted for up to 7 days without morphological and functional changes in CMs [49]. Zaitseva et al. designed hepatocyte growth factor (HGF) modRNA in incorporated nanofibrillar scaffolds, which allowed modRNA release from nanoscaffolds in a transient controlled way [50]. Lipid nanoparticles (LNPs), a lipid cargo that possesses a homogeneous lipid core against mRNA from extracellular ribonucleases, have been shown to help with intracellular mRNA transport [51]. In MI treatment, LNPs can effectively deliver modRNA to an injured heart, which stands as an excellent prospect for modRNA-based MI therapy. Through detecting the biodistribution of fluorescent-label LNPs, researchers have found that the majority of LNPs are distributed in heart fibroblasts in the infarct zone, but there are still a few LNPs in CMs and macrophages [52]. Moreover, the translated efficiency of formulated lipidoid nanoparticles (FLNPs) has been shown to be superior to other vehicles [53,54]. Paradoxically, researchers have also found that nanoparticle-encapsulated modRNA has lower translation efficiency than sucrose-citrate buffer-encapsulated modRNA, in which encoded protein can be detected within 10 min [35,55]. Interestingly, polymeric nanoparticles are novel and potential vectors for modRNA transportation with highly efficient transfection and low toxicity [56].

### 3.3. Delivery Method for modRNA

In addition to various materials, there are also different delivery methods for modRNA-based treatment. The two major methods for modRNA administration are intramyocardial and intravenous injections [57]. Multiple previous studies have indicated that intramyocardial injection of stem cells is more helpful in the recovery of heart function [58] since injected cells did not distribute throughout the whole body [59]. However, intramyocardial injection is still an invasive method, which can produce damage in the epicardial area and ventricular wall [60]. Intravenous injections can inhibit prolonged inflammatory processes compared to intramyocardial injections and have the ability of repeated injections several times [61].

Appropriate vehicles and injection methods are crucial to delivering disease-specific genes to the myocardium in cases of damage. A combination of gene modification, delivery materials, and methods to maximize the potential utility of modRNA for gene therapy could be considered in the future.

## 4. Modified mRNA-Based Therapy in MI Treatment

The limited ability of heart regeneration results in undesired morbidity and mortality after an MI. Factors such as inflammation, cardiac tissue remodeling, and the fibrotic environment contribute to limited CM proliferative activity after MI [62]. Stem cells are a group of unspecialized cells that have a special capacity to renew themselves and differentiate into other cell types [63]. Stem cells are associated with the repair of cardiac tissues mostly via direct differentiation into CMs, differentiation into endothelial cells, and secreting various trophic and paracrine factors, as well as inhibiting immune responses [64]. However, the ability of stem cells to differentiate into CMs is still disputable. Despite promising efficacy in preclinical and clinical studies, there are still some limitations that need to be addressed before broad clinical application of stem cell therapy, especially the extremely low survival rate, limited differentiation ability, safety concerns, as well as ethical issues [65]. Furthermore, an inflammatory microenvironment is also the main element to hinder the efficacy of stem cell therapy in cases of MI [66]. Thus, multiple technologies have been used to solve these challenges, therein, genetic manipulations regarding modRNA are also utilized in MI treatment. The favorable characteristics of modRNA as a transport vehicle, including highly efficient protein expression and flexible time dynamics, render it a potent choice for augmenting the regenerative ability of the myocardium. The protein can be detected within 3 h after injection of modRNA into the left ventricle, reaches a peak at 18 h post injection, and then gradually decreases in 6 days [67]. These time points correspond to the timeline of the MI process, i.e., CM death within 1 h after obstruction, secretion of proinflammatory factors at 4 h, fibrosis and angiogenesis after 2 days, and eventually, scar formation 2–3 weeks post MI [68]. Modified mRNA-based therapy with appropriate therapeutical genes can potentially treat MI by directly promoting proliferation and differentiation and inhibiting apoptosis of CMs, as well as enhancing paracrine effects in terms of promoting angiogenesis and inhibiting fibrosis of the cardiac microenvironment (Figure 2). Furthermore, apart from pathological processes, there are still multiple cell signaling pathways involved in mediating various cells after an MI, including CMs, endothelial cells, fibroblasts, monocytes, as well as stem cells [69]. These signaling pathways, for example, the PI3K/Akt, Notch, TGF-β/SMADs, Wnt/β-catenin, NLRP3/caspase-1, TLR4/MyD88/NF-κB, Nrf2/HO-1, RhoA/ROCK, MAPK, JAK/STAT, Hippo/YAP, and Sonic hedgehog pathways, mainly focus on several processes, including inflammation, oxidative stress, fibrosis, hypertrophy, apoptosis, survival, angiogenesis, and regeneration after an MI [69]. Next, we summarize some related genes for modRNA therapy in MI treatment published so far.

In terms of promoting the proliferation of CMs, modRNA delivery follistatin-like 1 (FSTL1) [39,70] or pyruvate kinase muscle isoenzyme 2 (PKM2) [43] both have this function. The transfer of N180Q mutant coupled with h-FSTL1 modRNA to the heart have been shown to trigger CM proliferation, reduce the infarct area, promote angiogenesis, and recover the heart function of MI mice [39]. Ajit et al. studied the role of *PKM2* in an MI mouse model by using bidirectionally regulated methods, losing *PKM2* via CM-specific *PKM2* knockout or gaining it via CM-specific PKM2 modRNA. The results indicated that PKM2 promoted proliferation and division during CM development but disappeared in adult mice [43]. The insulin-like growth factor-1 (IGF1) modRNA, encapsulated by the polyethyleneimine nanoparticle, protected CM survival and abrogated CM apoptosis in the scar border area. The cytoprotective effect of IGF1 was induced by activation of the Akt/the Erk pathway and the production of miR-1 and miR-133 [71]. Additionally, IGF-1 modRNA also promoted differentiation of the epicardial progenitor cells (EPCs) into adipose cells [72]. Hadas et al. performed a transcriptome, sphingolipid, and protein analysis to study the roles of sphingolipid metabolism in MI. Acid ceramidase (AC)-induced modRNA altered components of immune cells (decreased neutrophils), alleviated cardiac function, and minimized the infarct area in MI mice [73]. Meanwhile, the transcriptional co-stimulator yes-associated protein (YAP) also decreased necrosis of CMs and neutrophil infiltration by inhibiting the TLR pathway, thus improving heart function in MI mice [74].

Several works have validated that direct injection of vascular endothelial growth factor (VEGF) and vascular endothelial growth factor-A (VEGF-A) modRNA to MI mice resulted in expanded capillary density and maturity and decreased scar size, promoted heart function, and improved survival. Pulse-like transfer of VEGF-A modRNA led to the mobilization of EPCs and governed EPC differentiation toward vascular cell populations [75]. Additionally, VEGF-A modRNA has been recognized as a cell fate switch for embryonic stem cell (ESC)-derived Isl1^+^- ECs [76]. In larger animal-mini pigs, VEGF-A modRNA has also contributed to improving heart function and promoting angiogenesis [55]. Transfected *VEGF* modRNA in iPSC-CMs eventually promoted survival rates of iPSC-CMs, which constructed a tight vascular network in the injection zone [77]. Combination multi-gene therapy represents a potent technology for MI treatment. The 7G-modRNA method combines four cardiac reprogramming genes, i.e., Gata4, Mef2c, Tbx5, and Hand2, together with three helper genes, i.e., TGFb, Wnt8a, and AC, to induce CM-like cells. The results have shown that 7G-modRNA induced 57% CM-like cells in vitro and 25% CM-like cells in vivo. Interestingly, 7G-modRNA was unable to produce CMs, it only markedly promoted pro-angiogenic mesenchymal stromal cell markers and transcription factors [78]. Moreover, the 7G-modRNA method has been attributed to angiogenesis in ischemic limb injury [78], as well as VEGF modRNA [79]. Type 2 phosphatidylinositol-5-phosphate-4-kinase-gamma (Pip4k2c), an mTORC1 regulator delivered by modRNA, significantly inhibited TGFβ1 by its N-terminal motif, thus inhibiting cardiac fibrosis [80]. Another method to increase the number of CMs is to reprogram cardiac fibroblasts directly into CMs by viral introduction of lineage core transcription factors such as Gata4, Hand2, Mef2c, and Tbx5 [60]. Nevertheless, despite utilizing distinct approaches to enhance the redifferentiation rate, the efficiency of this method remains low, because of the limited number of fibroblast cells and eventual totally reprogrammed CMs [81]. Moreover, fibroblasts can be differentiated into endothelial cells, which can assistant the beating of CMs. Until now, there has been no relevant modRNA therapy of fibroblast differentiation for MI treatment [82]. As an emerging field of MI treatment, there should be a focus on more gene modification therapies.

The above-described studies reveal the efficiency of using direct or indirect methods for the treatment of MI. Modified mRNA-based therapy mainly focuses on the proliferation and differentiation of CMs; modRNA-based therapy can promote angiogenesis, promote stem cells to differentiate to ECs, or inhibit hostile microenvironment formation, such as fibrosis and hypoxia (Table 1).

## 5. Limitations and Future Directions of modRNA-Based Therapy in MI Treatment

Modified mRNA is a potent technique for MI treatment because it circumvents the limitations proposed by traditional DNA and protein-based therapies. Transfection by modRNA is attributed to transient protein expression, which no longer needs long-term protein expression. Nevertheless, there are still some problems that need to be addressed.

For modRNA to be functional in the body, it must be delivered to the specific scar area first, and then transported into the specific cells, evading the endosomal entrapment, and going to the cytosol, eventually translating to an encoded protein [86]. Any obstruction across the complicated procedure will significantly affect the final translation. However, modRNA dissolved in a solution is non-specifically absorbed or has premature clearance, which fails in the specific area at the first step [85]. Importantly, current modRNA-based therapy has no tissue and cell type-specific target ability in vivo, whereas adeno-associated virus vehicles can include tissue-specific promoters [87]. Moreover, the merit of modRNA, a short and transient expression of mRNA, also seems like a shortcoming. Whether the short-term expression is enough to induce authentic efficiency of myocardial regeneration or not, is still under debate [84]. Additionally, there are still some issues that have not been addressed, such as the optimal delivery route with an atraumatic operation (intramyocardial or transvascular), and the minimal effective dosage for cost-saving modRNA. To address these limitations, developing modRNA with long-term duration (2–3 weeks) or developing a method to repeat transfection of modRNA with non-invasion to sustain an effective protein level for a longer, under-controlled period, may accomplish a longer lasting efficacy. Moreover, modRNA target-specific genes or tissues are required because activating intracellular genes in incorrect cells are harmful. Combinations of genes or materials or other vectors are also needed to guarantee specific and non-invasive transport of RNA. More recently, application of the CRISPR-Cas13 system, which adopts bacteria to degrade viral RNA with high efficiency and to reduce the off-target effect, makes it possible to modify mRNA in a more efficient and safe way [88].

The clinical evaluation of modRNA-based therapy for MI is still in the initial state. In 2019, a phase I trial was conducted to evaluate the therapeutic effect and safety of AZD8601, an experimental VEGF-A-mRNA, which was formulated in biocompatible citrate-buffered saline and optimized for high-efficiency VEGF-A expression with minimal innate immune response [89]. Directly injecting a novel medicine into a human heart is extremely dangerous; therefore, this phase I safety study injected VEGF-A mRNA into the skin of 33 men, and showed temporary and plentiful production of a therapeutic protein without severe side effects. In 2021, positive results were reported from a phase 2 study that evaluated the clinical effect of AZD8601 using epicardial injection in patients who underwent coronary artery bypass grafting with moderately decreased left ventricular function (ejection fraction 30–50%) [90]. The results showed that exploratory efficacy endpoints were met, including increased left ventricular ejection fraction and decreased NT-proBNP level. Although there are still limited results regarding modRNA-MI therapy, the early clinical trials presented today are a result of pushing new boundaries in the treatment of cardiovascular and other ischemic vascular diseases to address serious unmet needs, with the goal of improving patients’ lives. Future advanced clinical trials incorporating more MI patients should be conducted within 10 years. Only in these ways can patients indeed get the benefits from the modRNA delivery system.

## 6. Conclusions

Despite numerous studies that have been undertaken in terms of MI treatment, there are still many obstacles to curing MI. Especially, CMs have a limited capacity to regenerate even by strongly extrinsic or intrinsic stimuli. Modified mRNA-based therapy is an excellent therapeutic method to solve preclinical and clinical questions for cardiac regeneration with its properties of efficiency, non-immunogenicity, transiency, and relative safety. However, so far, a comprehensive summary of modRNA-based therapy has not been proposed, and further exploration of modRNA-based therapy needs to be discussed.

Firstly, we summarized the optimization of modRNA-based therapy, including gene modification and delivery materials of modRNA. In the articles discussed above, the translation efficiency of modRNA-based therapy can be further improved via editing nucleotides or a combination of modified nucleosides, optimization of the 5′ UTR and 3′ UTR vector sequences, optimal pH value, as well as optimal dosage values in vitro and in vivo. Additionally, appropriate vehicles, such as RNAiMAX in vitro, nanoparticles, LNPs, and FLNPs in vivo, are crucial to delivering disease-specific genes to the myocardium. A combination of gene modification and delivery materials, can maximize the potential utility of modRNA for gene therapy.

Secondly, we focused on the efficacy and mechanisms of modRNA-based therapy in MI treatment; modRNA-based therapy can promote the proliferation of CMs and differentiation of stem cells and can also alleviate the harsh microenvironment of the myocardium, for example, inhibiting fibrosis and promoting angiogenesis. Therein, VEGF and VEGF-A both have important effects on angiogenesis and pro-angiogenic differentiation in modRNA-based therapy. Other genes, such as FSTL1 and Pkm2, can promote CM proliferation, meanwhile, IGF-1, a growth factor activity, and integrin binding related gene, can promote differentiation of EPCs. Multiple genes-combined modRNA termed cocktail therapy, markedly upregulate pro-angiogenic MSC markers and transcription factor.

Finally, we summarized the principal obstacles and future direction of modRNA-based therapy. There is a need to develop modRNA with more safety, cost-effectiveness, stable transfer vectors, and relatively long-term controlled protein expression. Finally, the scale-up from animal experiments and clinical translation requires non-invasive methods. The clinical translation should be evaluated as soon as there is adequate evidence from animal studies.

## Figures and Tables

**Figure 1 ijms-24-04737-f001:**
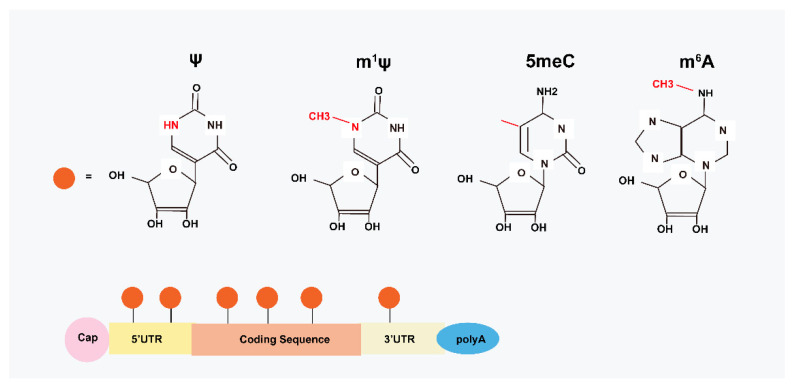
Structural basis of modRNA. Modified mRNA-based therapy by chemically modified nucleotides, including insertion of methyl or hydroxylate groups and replacement of uridine with ψ. For example, 5meC, m^6^A, and m^1^ψ nucleotides are commonly used in MI treatment.

**Figure 2 ijms-24-04737-f002:**
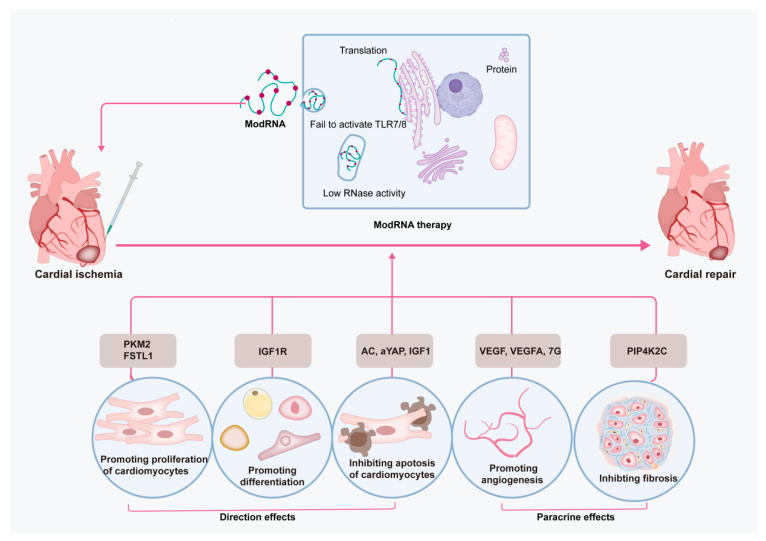
The mechanisms of modRNA-based therapy in MI treatment. modRNA-based therapy not only directly protects CMs but also through the paracrine effect alleviates the harsh microenvironment of the myocardium, for example, inhibiting fibrosis and promoting angiogenesis. Therein, VEGF and VEGF-A both have important effects on angiogenesis and pro-angiogenic differentiation in modRNA-based therapy. FSTL1 and PKM2 modRNA promote CM proliferation, and IGF-1R modRNA promotes differentiation of EPDCs. AC, aYAP, and IGF1 modRNA inhibit apoptosis of CMs. PIP4K2C modRNA inhibits MI fibrosis.

**Table 1 ijms-24-04737-t001:** Mechanisms of the modRNA therapeutic strategy for MI treatment.

Functions	Genes	Delivery Materials	Delivery Methods	Cells	Animals	Effects
Promoting proliferation	FSTL1	Sucrose-citrate buffer	IM	CMs	Mice	Increased cardiac function, decreased scar size, and increased capillary density [39]
	PKM2	Sucrose-citrate buffer	IM	CMs	Mice	Enhanced cardiac function and improved long-term survival [43]
Promoting differentiation	IGF1	RNAiMAX	IM	EPCs	Mice	Governed epicardial adipose tissue formation in the context of myocardial injury by redirecting the fate of Wt1^+^ lineage cells [72]
Promoting angiogenesis	VEGFA	RNAiMAX	IM	EPCs	Mice	Improved heart function and enhanced long-term survival [75]
	VEGF	Sucrose-citrate buffer	IM	iPSC-CMs	Rats	Improved heart function and enhanced long-term survival of recipients [77]
	VEGFA	Citrate-saline	IM	CMs	Pigs, mice	Improved systolic ventricular function and limited myocardial damage, left ventricular ejection fraction, border zone arteriolar and capillary density increased, and myocardial fibrosis decreased [55]
	VEGFA	RNAiMAX	IM	Isl1^+^ progenitors	Mice	Driven endothelial specification, engraftment, and survival following transplantation [76]
	7G	Sucrose-citrate buffer	IM	MSCs	Mice	Improved cardiac function, scar size, long-term survival, and capillary density [78]
Inhibiting fibrosis	PIP4K2C	Sucrose-citrate buffer	IM	CMs, fibroblasts	Mice	Attenuating cardiac hypertrophy and fibrosis, enhanced long-term survival
Promoting survival	AC	RNAiMAX	IM	CMs	Mice	Improved heart function, longer survival, and reduced scar size [73]
	aYAP	Saline	IM	CMs	Mice	Improved heart function and suppressed cardiac hypertrophy [74]
	IGF-1	Polyethylenimine-based nanoparticle	IM	CMs	Mice	Promoted CM survival and abrogated cell apoptosis under hypoxia-induced apoptosis conditions [83]
Promoting delivery efficiency	-	LNP	i.v. via the tail vein	Fibroblasts	Mice	Most targeted cells were cardiac fibroblasts but also some CMs and macrophages [52]
	eGFP	FLNP	IM or intracoronary administration	-	Rats, pigs	Increasing modRNA expression in heart [53]
	GEVIs, GECIs	Lipofectamine stem transfection reagent	In vitro	hiPSC-CMs	-	Delivered strong and stable signals in hiPSC-CMs [40]
	Varies	RNAiMAX	IM	-	Mice	Effective synthesis of modRNA for in vivo use [84]
	Luciferase	Sucrose-citrate buffer	IM	-	Mice	Optimized modRNA amount, time, and delivery [35]
	Luciferase	Sucrose-citrate buffer	IM	-	Mice	Increased translation by replacing 5′ UTR [16]
	eGFP	RNAiMAX	In vitro	-	-	Improved in vitro transcription [26]
	EGFP, mCherry, Fluc	Alginate, nanomater	IM	-	Pigs	Optimized M3RNA delivery into myocardium [49]
	HGF	Nanofibrillar scaffolds	IM	Fibroblasts	Mice	Improved translation efficiency [50]
	GFP, luciferase	Sucrose-citrate buffer	In vitro	CMs	-	Improved modRNA yield and translation efficiency, reduced its immunogenicity [85]

## Data Availability

The data that support the findings of this study are available from the corresponding author upon reasonable request.

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
