# Peer review of "Modified mRNA as a Treatment for Myocardial Infarction"

_ijms, 2023, doi:10.3390/ijms24054737_

Round 1
Reviewer 1 Report
Generally, the topic is of interest and the manuscript is of fluid reading. However, I think it could be improved. Therefore, I would like to recommend that:
1. Please check for plagiarism.
2. Please carefully revise gene/protein nomenclature.
3. The abstract could benefit if the authors address their main achievements and what can be ameliorated in the future and how it could be done.
4. Please address some spelling questions. For example, lines 37, 47, 84, 125, 182-193.
5. It will be prouder if you use among others than “and so on” (e.g., line 87)
6. Figure 1 (page 5) is far from its first mention (page 2).
7. The section “Delivery materials and methods” could be improved. You can make a difference by adding information about the potential use of non-viral vs. viral vectors and including a figure with the different nanosystems. Please look at the following manuscripts:
https://www.mdpi.com/1999-4923/14/3/512
https://www.ncbi.nlm.nih.gov/pmc/articles/PMC5113109/
https://www.ncbi.nlm.nih.gov/pmc/articles/PMC5113109/
https://www.mdpi.com/1422-0067/23/24/15514
https://www.frontiersin.org/articles/10.3389/fphar.2021.722728/full
https://www.nature.com/articles/s41573-021-00355-6
Maybe the title of this subsection is Delivery vectors for modRNA?
8. Is there clinical trials using this kind of approach?
9. What about programmable RNA? (https://www.frontiersin.org/articles/10.3389/fgene.2022.834413/full)
Could you please discuss it?
Author Response
We truly appreciate for the careful checking for our grammar mistakes and insightful comments. Our point-by-point responses are listed as follows.
Comment 1: Please check for plagiarism.
Response 1: We used turnitin to do plagiarism, the duplicated rate is 8%.
Comment 2: Please carefully revise gene/protein nomenclature.
Response 2: Thanks, we carefully revised our nonmenclature, please refer to our revised manuscript.
Comment 3: The abstract could benefit if the authors address their main achievements and what can be ameliorated in the future and how it could be done.
Response 3: We revised our abstract as you suggested.
Comment 4: Please address some spelling questions. For example, lines 37, 47, 84, 125, 182-193.
Response 4: We revised our spelling.
Comment 5: It will be prouder if you use among others than “and so on” (e.g., line 87)
Response 5: We revised as you suggested.
Comment 6: Figure 1 (page 5) is far from its first mention (page 2).
Response 6: We splited Figure1 to two parts, please refer to our revised manuscript.
Comment 7: The section “Delivery materials and methods” could be improved. You can make a difference by adding information about the potential use of non-viral vs. viral vectors and including a figure with the different nanosystems. Please look at the following manuscripts:
https://www.mdpi.com/1999-4923/14/3/512
https://www.ncbi.nlm.nih.gov/pmc/articles/PMC5113109/
https://www.ncbi.nlm.nih.gov/pmc/articles/PMC5113109/
https://www.mdpi.com/1422-0067/23/24/15514
https://www.frontiersin.org/articles/10.3389/fphar.2021.722728/full
https://www.nature.com/articles/s41573-021-00355-6
Maybe the title of this subsection is Delivery vectors for modRNA?
Response 7: We revised as you suggested. However, we only focus on modRNA therapy in myocardial infarction, so in this area we only find references about lipid nanoparticle. It’s far from to build a figure of nanosystems. We will consider this valuable advice in another review focusing on delivery vectors for RNA. Thank you so much.
Comment 8: Is there clinical trials using this kind of approach?
Response 8: There is one new medine-AZD8601, which encoded VEGF-A mRNA, has already used in clinical trial. We discussed it in line 379-396. Please refer to our revised manuscript.
Comment 9: What about programmable RNA (https://www.frontiersin.org/articles/10.3389/fgene.2022.834413/full) Could you please discuss it?
Response 9: We discussed it at line 376-378.
Reviewer 2 Report
The manuscript is very interesting and I consider that its publication is relevant. However, I have some observations.
Although the advantages of modRNA over a conventional RNA are clearly mentioned, it would be very relevant if they mention any previous work that shows the difference that exists in effectiveness or level of protein production between a modRNA and an identical unmodified RNA sequence. This is to be able to measure how much difference exists between both vectors due solely to the modifications in their nucleotides.
It would be advisable to make a section on the advantages that modRNA has over other more widely used strategies, such as the use of adenoviral vectors.
Show some application already in humans of a modRNA (clinical trial in any disease), mentioning its results to demonstrate the efficacy and feasibility of this strategy for its future use in humans in myocardial infarctions.
Finally, it would be very useful for readers who are not experts in the area, if the authors estimated the time in which they believe that these technologies will pass to the experimental phase in humans for IM, or perhaps an approximate to be able to have them commercially for the first time in some country. I know this is hard to predict, but at least it has to be said that it will be many years before they are available to the general public. Some comment in this regard would be necessary so that doctors or patients do not have false expectations regarding its use in the near future.
Author Response
We truly appreciate for the positive and insightful comments. Our point-by-point responses are listed as follows.
Comment 1: Although the advantages of modRNA over a conventional RNA are clearly mentioned, it would be very relevant if they mention any previous work that shows the difference that exists in effectiveness or level of protein production between a modRNA and an identical unmodified RNA sequence. This is to be able to measure how much difference exists between both vectors due solely to the modifications in their nucleotides.
Response 1: The comparison of translate efficiency between a modRNA and unmodified RNA was referred in line -107-116.
Comment 2: It would be advisable to make a section on the advantages that modRNA has over other more widely used strategies, such as the use of adenoviral vectors.
Response 2: We discussed the advantage of modRNA over other strategies in line 147-181. Please refer to our revised manuscript.
Comment 3: Show some application already in humans of a modRNA (clinical trial in any disease), mentioning its results to demonstrate the efficacy and feasibility of this strategy for its future use in humans in myocardial infarctions.
Response 3: Although there are some clinicals trial in other diseases like cancer or pneumonia, we still want to find clinical support from cardiovascular disease. There is one new medine-AZD8601, which encoded VEGF-A mRNA, has already used in clinical trial. We discussed it in line 379-396. Please refer to our revised manuscript.
Comment 4: Finally, it would be very useful for readers who are not experts in the area, if the authors estimated the time in which they believe that these technologies will pass to the experimental phase in humans for IM, or perhaps an approximate to be able to have them commercially for the first time in some country. I know this is hard to predict, but at least it has to be said that it will be many years before they are available to the general public. Some comment in this regard would be necessary so that doctors or patients do not have false expectations regarding its use in the near future.
Response 4: Because these is two early phase clinical trials already have already got the positive result, we think the treatment of modRNA therapy in human cardiovascular disease will undergone within next decade. We discussed it in line 394-396.